# Don't stack layers in graph neural networks, wire them randomly

## Abstract

Graph neural networks have become a staple in problems addressing learning and analysis of data defined over graphs. However, several results suggest an inherent difficulty in extracting better performance by increasing the number of layers. Besides the classic vanishing gradient issues, recent works attribute this to a phenomenon peculiar to the extraction of node features in graph-based tasks, i.e., the need to consider multiple neighborhood sizes at the same time and adaptively tune them. In this paper, we investigate the recently proposed randomly wired architectures in the context of graph neural networks. Instead of building deeper networks by stacking many layers, we prove that employing a randomly-wired architecture can be a more effective way to increase the capacity of the network and obtain richer representations. We show that such architectures behave like an ensemble of paths, which are able to merge contributions from receptive fields of varied size. Moreover, these receptive fields can also be modulated to be wider or narrower through the trainable weights over the paths. We also provide extensive experimental evidence of the superior performance of randomly wired architectures over three tasks and five graph convolution definitions, using a recent benchmarking framework that addresses the reliability of previous testing methodologies.

## 1 Introduction

Data defined over the nodes of graphs are ubiquitous. Social network profiles (Hamilton et al., 2017), molecular interactions (Duvenaud et al., 2015), citation networks (Sen et al., 2008), 3D point clouds (Simonovsky & Komodakis, 2017) are just examples of a wide variety of data types where describing the domain as a graph allows to encode constraints and patterns among the data points. Exploiting the graph structure is crucial in order to extract powerful representations of the data. However, this is not a trivial task and only recently graph neural networks (GNNs) have started showing promising approaches to the problem. GNNs (Wu et al., 2020) extend the deep learning toolbox to deal with the irregularity of the graph domain. Much of the work has been focused on defining a graph convolution operation (Bronstein et al., 2017), i.e., a layer that is well-defined over the graph domain but also retains some of the key properties of convolution such as weight reuse and locality. A wide variety of such graph convolution operators has been defined over the years, mostly based on neighborhood aggregation schemes where the features of a node are transformed by processing the features of its neighbors. Such schemes have been shown to be as powerful as the Weisfeiler-Lehman graph isomorphism test (Weisfeiler & Lehman, 1968; Xu et al., 2019), enabling them to simultaneuosly learn data features and graph topology.

However, contrary to classic literature on CNNs, few works (Li et al., 2019a; Dehmamy et al., 2019; Xu et al., 2018; Dwivedi et al., 2020) addressed GNNs architectures and their role in extracting powerful representations. Several works, starting with the early GCN (Kipf & Welling, 2017), noticed an inability to build deep GNNs, often resulting in worse performance than that of methods that disregard the graph domain, when trying to build anything but very shallow networks. This calls for exploring whether advances on CNN architectures can be translated to the GNN space, while understanding the potentially different needs of graph representation learning.

Li et al. (2019b) suggest that GCNs suffer from oversmoothing as several layers are stacked, resulting in the extraction of mostly low-frequency features. This is related to the lack of self-loop information in this specific graph convolution. It is suggested that ResNet-like architectures mitigate the problem as the skip connections supply high frequency contributions. Xu et al. (2018) point out that the size of the receptive field of a node, i.e., which nodes contribute to the features of the node under

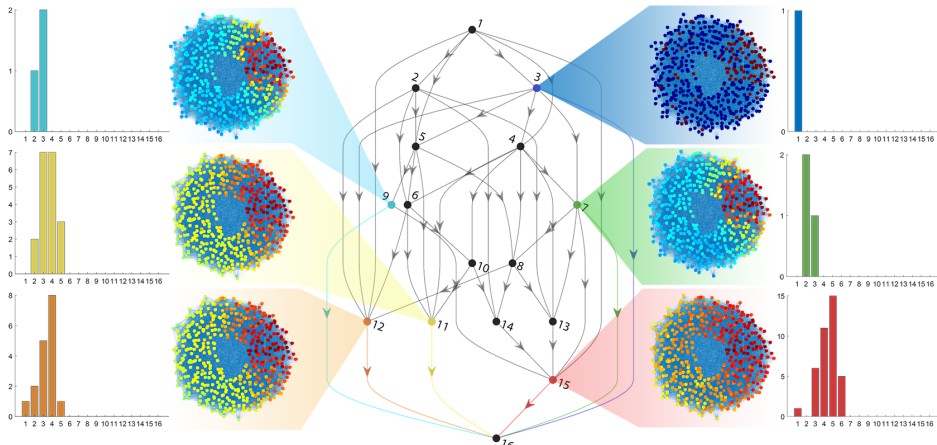

Figure 1: Random architectures aggregate ensembles of paths. This creates a variety of receptive fields (effective neighborhood sizes on the domain graph) that are combined to compute the output. Figure shows the domain graph where nodes are colored (red means high weight, blue low weight) according to the receptive field weighted by the path distribution of a domain node. The receptive field is shown at all the architecture nodes directly contributing to the output. Histograms represent the distribution of path lengths from source to architecture node.

consideration, plays a crucial role, but it can vary widely depending on the graph and too large receptive fields may actually harm performance. They conclude that for graph-based problems it would be optimal to learn how to adaptively merge contributions from receptive fields of multiple size. For this reason they propose an architecture where each layer has a skip connection to the output so that contributions at multiple depths (hence sizes of receptive fields) can be merged. Nonetheless, the problem of finding methods for effectively increasing the capacity of graph neural networks is still standing, since stacking many layers has been proven to provide limited improvements (Li et al., 2019b; Oono & Suzuki, 2019; Alon & Yahav, 2020; NT & Maehara, 2019).

In this paper, we argue that the recently proposed randomly wired architectures (Xie et al., 2019) are ideal for GNNs. In a randomly wired architecture, "layers" are arranged according to a random directed acyclic graph and data are propagated through the paths towards the output. Such architecture is ideal for GNNs because it realizes the intuition of Xu et al. (2018) of being able of merging receptive fields of varied size. Indeed, the randomly wired network can be seen as an extreme generalization of their jumping network approach where layer outputs can not only jump to the network output but to other layers as well, continuously merging receptive fields. Hence, randomly wired architectures provide a way of effectively scaling up GNNs, mitigating the depth problem and creating richer representations. Fig. 1 shows a graphical representation of this concept by highlighting the six layers directly contributing to the output, having different receptive fields induced by the distribution of paths from the input.

Our novel contributions can be summarized as follows: i) we are the first to analyze randomly wired architectures and show that they are generalizations of ResNets when looked at as ensembles of paths (Veit et al., 2016); ii) we show that path ensembling allows to merge receptive fields of varied size and that it can do so *adaptively*, i.e., trainable weights on the architecture edges can tune the desired size of the receptive fields to be merged to achieve an optimal configuration for the problem; iii) we introduce improvements to the basic design of randomly wired architectures by optionally embedding a path that sequentially goes through all layers in order to promote larger receptive fields when needed, and by presenting MonteCarlo DropPath, which decorrelates path contributions by randomly dropping architecture edges; iv) we provide extensive experimental evidence, using a recently introduced benchmarking framework (Dwivedi et al., 2020) to ensure significance and reproducibility, that randomly wired architectures consistently outperform ResNets, often by large margins, for five of the most popular graph convolution definitions on three different tasks.

## 2 BACKGROUND

### 2.1 GRAPH NEURAL NETWORKS

A major shortcoming of CNNs is that they are unable to process data defined on irregular domains. In particular, one case that is drawing attention is when the data structure can be described by a graph and the data are defined as vectors on the graph nodes. This setting can be found in many applications, including 3D point clouds (Wang et al., 2019; Valsesia et al., 2019), computational biology (Alipanahi et al., 2015; Duvenaud et al., 2015), and social networks (Kipf & Welling, 2017). However, extending CNNs from data with a regular structure, such as images and video, to graph-structured data is not straightforward if one wants to preserve useful properties such as locality and weight reuse.

GNNs redefine the convolution operation so that the new layer definition can be used on domains described by graphs. The most widely adopted graph convolutions in the literature rely on message passing, where a weighted aggregation of the feature vectors in a neighborhood is computed. The GCN (Kipf & Welling, 2017) is arguably the simplest definition, applying the same linear transformation to all the node features, followed by neighborhood aggregation and non-linear activation:

$$\mathbf{h}_i^{(l+1)} = \sigma\left(\frac{1}{|\mathcal{N}_i|}\sum_{j\in\mathcal{N}_i}\mathbf{W}\mathbf{h}_j^{(l)}\right).$$

Variants of this definition have been developed, e.g., GraphSage (Hamilton et al., 2017) concatenates the feature vector of node $i$ to the feature vectors of its neighbors, so that self-information can also be exploited; GIN (Xu et al., 2019) uses a multilayer perceptron instead of a linear transform, replaces average with sum to ensure injectivity and proposes a different way of computing the output by using all the feature vectors produced by the intermediate layers. These definitions are all isotropic because they treat every edge in the same way. It has been observed that better representation capacity can be achieved using anistropic definitions, where every edge can have a different transformation, at the cost of increased computational complexity. The Gated GCN (Bresson & Laurent, 2017) and GAT (Veličković et al., 2017) definitions fall in this category.

### 2.2 RANDOMLY WIRED ARCHITECTURES

In recent work, Xie et al. (2019) explore whether it is possible to avoid handcrafted design of neural network architectures and, at the same time, avoid expensive neural architecture search methods (Elsken et al., 2019), by designing random architecture generators. They show that "layers" performing convolution, normalization and non-linear activation can be connected in a random architecture graph. Strong performance is observed on the traditional image classification task by outperforming state-of-the-art architectures. The authors conjecture that random architectures generalize ResNets and similar constructions, but the underlying principles of their excellent performance are unclear, as well as whether the performance translates to tasks other than image recognition or to operations other than convolution on grids.

## 3 RANDOMLY WIRED GNNS

In this section, we first introduce randomly wired architectures and the notation we are going to use. We then analyze their behavior when viewed as ensembles of paths.

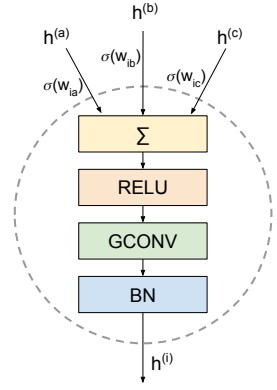

A randomly wired architecture consists of a directed acyclic graph (DAG) connecting a source architecture node, which is fed with the input data, to a sink architecture node. One should not confuse the architecture DAG with the graph representing the GNN domain: to avoid any source of confusion we will use the terms *architecture nodes* (edges) and *domain nodes* (edges), respectively. A domain node is a node of the graph that is fed as input to the GNN. An architecture node is effectively a GNN layer performing the following operations (Fig. 2): i) aggregation of the inputs from other architecture nodes via a weighted sum as in (Xie et al., 2019):

$$\mathbf{h}^{(i)} = \sum_{j\in\mathcal{A}_i}\omega_{ij}\mathbf{h}^{(j)} = \sum_{j\in\mathcal{A}_i}\sigma(w_{ij})\mathbf{h}^{(j)}, \quad i = 1,...,L-1 \quad (1)$$

Figure 2: An architecture node is equivalent to a GNN layer.

being $\sigma$ a sigmoid function, $\mathcal{A}_i$ the set of direct predecessors of the architecture node $i$, and $w_{ij}$ a scalar trainable weight; ii) a non-linear activation; iii) a graph-convolution operation (without output activation); iv) batch normalization.

The architecture DAG is generated using a random graph generator. In this paper, we will focus on the Erdős-Renyi model where the adjacency matrix of the DAG is a strictly upper triangular matrix with entries being realizations of a Bernoulli random variable with probability $p$. If multiple input architecture nodes are randomly generated, they are all wired to a single global input. Multiple output architecture nodes are averaged to obtain a global output. Other random generators may be used, e.g., small-world and scale-free random networks have been studied in (Xie et al., 2019). However, a different generator will display a different behavior concerning the properties we study in Sec. 3.1.

### 3.1 RANDOMLY WIRED ARCHITECTURES BEHAVE LIKE PATH ENSEMBLES

It has already been shown that ResNets behave like ensembles of relatively shallow networks, where one can see the ResNet architecture as a collection of paths of varied lengths (Veit et al., 2016). More specifically, in a ResNet with $n$ layers, where all layers have a skip connection except the first one and the last one, there are exactly $2^{L-2}$ paths, whose lengths follow a Binomial distribution (i.e., the number of paths of length $l$ from layer $k$ to the last layer is $\binom{L-k-1}{l-2}$), and the average path length is $\frac{L}{2}+1$ (Veit et al., 2016). In this section, we show that a randomly wired neural network can also be considered as an ensemble of networks with varied depth. However, in this case, the distribution of the path length is different from the one obtained with the ResNet, as shown in the following lemma (proof in the supplementary material).

**Lemma 3.1.** *Let us consider a randomly wired network with $L$ architecture nodes, where the architecture DAG is generated according to a Erdős-Renyi graph generator with probability $p$. The average number of paths of length $l$ from node $k$ to the sink, where $k < L$, is $\mathbb{E}[N_l^{(k)}] = \binom{L-k-1}{l-2} p^{l-1}$ and the average total number of paths from node $k$ to the sink is $\mathbb{E}[N^{(k)}] = p(1+p)^{L-k-1}$.*

We can observe that if $p = 1$, the randomly wired network converges to the ResNet architecture. This allows to think of randomly wired architectures as generalizations of ResNets as they enable increased flexibility in the number and distribution of paths instead of enforcing the use of all $2^{L-2}$.

### 3.2 RECEPTIVE FIELD ANALYSIS

In the case of GNNs, we define the receptive field of a domain node as the neighborhood that affects the output features of that node. As discussed in Sec. 1, the work in (Xu et al., 2018) highlights that one of the possible causes of the depth problem in GNNs is that the size of the receptive field is not adaptive and may rapidly become excessively large. Inspired by this observation, in this section we analyze the receptive field of a randomly wired neural network. We show that the receptive field of the output is a combination of the receptive fields of shallower networks, induced by each of the paths. This allows to effectively merge the contributions from receptive fields of varied size. Moreover, we show that the trainable parameters along the path edges modulate the contributions of various path lengths and enable adaptive receptive fields, that can be tuned by the training procedure.

We first introduce a definition of the receptive field of a feedforward graph neural network[1].

**Definition 3.1.** *Given a feedforward graph neural network with $L$ layers, the receptive field of radius $L$ of a domain node is its $L$-hop neighborhood.*

In a randomly wired architecture, each path induces a corresponding receptive field whose radius depends on the length of the path. Then, the receptive field at the output of the network is obtained by combining the receptive fields of all the paths. In order to analyze the contribution of paths of different lengths to the receptive field of the network, we introduce the concept of distribution of the receptive field radius of the paths. Notice that if we consider a feedforward network with $L$ layers, the distribution of the receptive field radius is a delta centered in $L$.

The following lemma allows to analyze the distribution of the receptive field radius in a randomly wired architecture.

---

[1]We use the term "feedforward neural network" to indicate an architecture made of a simple line graph, without skip connections: this is a representation of one path.

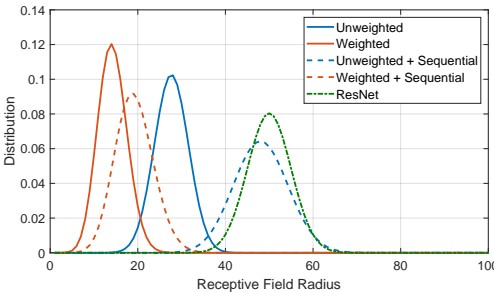 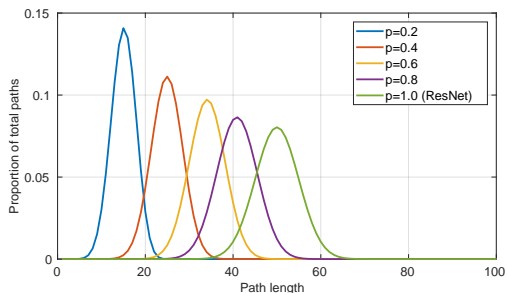

Figure 3: Distribution of receptive field radius ($p = 0.4$, $\omega_{ij} = 1$ for unweighted, $\omega_{ij} = 0.5$ for weighted).

Figure 4: Path distribution as function of architecture edge probability.

**Lemma 3.2.** *The derivative $\frac{\partial y}{\partial x_0}$ of the output $y$ of a randomly wired architecture with respect to the input $x_0$ is*

$$\frac{\partial y}{\partial x_0} = \sum_{p \in \mathcal{P}} \frac{\partial y_p}{\partial x_0} = \sum_{p \in \mathcal{P}} \prod_{\{i,j\} \in \mathcal{E}^p} \omega_{ij} \frac{\partial \bar{y}_p}{\partial x_0} = \sum_{l=2}^{L} \sum_{p \in \mathcal{P}^l} \lambda_p \frac{\partial \bar{y}_p}{\partial x_0},$$

*where $y_p$ is the output of path $p$, $\bar{y}_p$ is the output of path $p$ when we consider all the aggregation weights equal to 1, $\lambda_p = \frac{\partial y_p}{\partial x_0} / \frac{\partial \bar{y}_p}{\partial x_0}$, $\mathcal{P}$ is the set of all paths from source to sink, $L$ is the number of architecture nodes, $\mathcal{P}^l$ is the set of paths from source to sink of length $l$ and $\mathcal{E}^p$ is the set of edges of the path $p$.*

*Proof.* Direct computation. $\qquad\square$

From Lemma 3.2, we can observe that the contribution of each path to the gradient is weighted by its corresponding architecture edge weights. Thus, we can define the following distribution $\rho$ of the receptive field radius:

$$\rho_l = \sum_{p \in \mathcal{P}^l} \lambda_p = \sum_{p \in \mathcal{P}^l} \prod_{\{i,j\} \in \mathcal{E}^p} \omega_{ij} \qquad \text{for } l = 2, ..., n, \tag{2}$$

where we have assumed that the gradient $\frac{\partial \bar{y}_p}{\partial x_0}$ depends only on the path length, as done in (Veit et al., 2016). This is a reasonable assumption if all the architecture nodes perform the same operation. The distribution of the receptive field radius is therefore influenced by the architecture edge weights. Figure 3 shows an example of how such weights can modify the radius distribution. If we consider $\omega_{ij} = 1$ for all $i$ and $j$, we obtain that the radius distribution is equal to the path length distribution. In order to provide some insight into the role of parameter $p$ in the distribution of the receptive field radius, we focus on this special case and analyze the distribution of the path lengths in a randomly wired architecture by introducing the following Lemma (proof in the supplementary material).

**Lemma 3.3.** *Let us consider a randomly wired network with $L$ architecture nodes, where the architecture DAG is generated according to a Erdős-Renyi graph generator with probability $p$. The average length of the paths from node $k$ to the sink is $\mathbb{E}[l^{(k)}] \approx \frac{p}{1+p}(L - k - 1) + 2$.*

Therefore, if $p = 1$ and $\omega_{ij} = 1$ for all $i$ and $j$ the radius distribution is a Binomial distribution centered in $\frac{L}{2} + 1$ (as in ResNets), instead when $p < 1$ the mean of the distribution is lower. The path length distribution for different $p$ values is shown in Fig. 4. This shows that, differently from feedforward networks, the receptive field of ResNets and randomly wired architectures is a combination of receptive fields of varied sizes, where most of the contribution is given by shallow paths, i.e. smaller receptive fields. The parameter $p$ of the randomly wired neural network influences the distribution of the receptive field radius: a lower $p$ value skews the distribution towards shallower paths, instead a higher $p$ value skews the distribution towards longer paths.

After having considered the special case where $\omega_{ij} = 1$ for all $i$ and $j$, we now focus on the general case. Since the edge architecture weights are trainable parameters, they can be adapted to optimize the distribution of the receptive field radius. This is one of the strongest advantages provided by randomly wired architectures with respect to ResNets. This is particularly relevant in the context of GNNs, where we may have a non-uniform growth of the receptive field caused by the irregularity

of the graph structure (Xu et al., 2018). Notice that the randomly wired architecture can be seen as a generalization of the jumping knowledge networks proposed in (Xu et al., 2018), where all the architecture nodes, not only the last one, merge contributions from previous nodes. We also remark that, even if we modify the ResNet architecture by adding trainable weights to each branch of the residual module, we cannot retrieve the behaviour of the randomly wired architecture. In fact, the latter has intrinsically more granularity than a ResNet: the expected number of architecture edge weights of a randomly wired network is $\frac{pL(L+1)}{2}$, instead a weighted ResNet has only $2(L-2)$ weights. Ideally, we would like to weight each path independently (i.e., directly optimizing the value of $\lambda_p$ in Eq. (3.2)). However, this is unfeasible because the number of parameters would become excessively high and the randomly wired architecture provides an effective tradeoff. Given an architecture node, weighting in a different way each input edge is important because to each edge corresponds a different length distribution of the paths going through such edge, as shown by the following Lemma (proof in supplementary material).

**Lemma 3.4.** *Let us consider a randomly wired network with $n$ architecture nodes, where the architecture DAG is generated according to a Erdős-Renyi graph generator with probability $p$. Given an edge $\{i, j\}$ between the architecture nodes $i$ and $j$ where $i < j$, the average length of the paths from the source to the sink going through that edge is $\mathbb{E}[l_{ij}] \approx \frac{p}{1+p}(L - (j - i) - 3) + 4$.*

### 3.3 SEQUENTIAL PATH

In the previous sections we have shown that a randomly wired architecture behaves like an ensemble of paths merging contribution from receptive fields of varied size, where most of the contribution is provided by shallow paths. As discussed previously, this provides numerous advantages with respect to feedforward networks and ResNets. However, some graph-based tasks may actually benefit from a larger receptive field (Li et al., 2019b), so it is interesting to provide randomly wired architectures with mechanisms to directly promote longer paths. Differently from ResNets, in a randomly wired neural network with $L$ architecture nodes the longest path may be shorter than $L$, leading to a smaller receptive field. In order to overcome this issue, we propose to modify the generation process of the random architecture by imposing that it should also include the sequential path, i.e., the path traversing all architecture nodes. This design of the architecture skews the initial path length distribution towards longer paths, which has the effect of promoting their usage. Nevertheless, the trainable architecture edge weights will ultimately define the importance of such contribution. Fig. 3 shows an example of how including the sequential path changes the distribution of the receptive field radius.

### 3.4 MONTECARLO DROPPATH REGULARIZATION

The randomly wired architecture offers new degrees of freedom to introduce regularization techniques. In particular, one could delete a few architecture edges during training with probability $p_{\text{drop}}$ as a way to avoid co-adaptation of architecture nodes. This is reminiscent of DropOut (Srivastava et al., 2014) and DropConnect (Wan et al., 2013), although it is carried out at a higher level of abstraction, i.e., connections between "layers" instead of neurons. It is also reminiscent of techniques used in Neural Architecture Search (Zoph et al., 2018) and the approach used in ImageNet experiments in (Xie et al., 2019), although implementation details are unclear for the latter.

We propose to use a MonteCarlo approach where paths are also dropped in testing. Inference is performed multiple times for different realizations of dropped architecture edges and results are averaged. This allows to sample from the full predictive distribution induced by DropPath, as in MonteCarlo DropOut (Gal & Ghahramani, 2015). It is worth noting that MonteCarlo DropPath decorrelates the contributions of paths in Eq. (3.2) even if they share architecture edges (proof in supplementary material), thus allowing finer control over the modulation of the receptive field radius.

### 4 EXPERIMENTAL RESULTS

Experimental evaluation of GNNs is a topic that has recently received great attention. The emerging consensus is that benchmarking methods routinely used in past literature are inadequate and lack reproducibility. In particular, Vignac et al. (2020) showed that commonly used citation network datasets like CORA, CITESEER, PUBMED are too simple and skew results towards simpler architectures or even promote ignoring the underlying graph. TU datasets are also recognized to be too small (Errica et al., 2019) and the high variability across splits does not allow for sound comparisons across methods. In order to evaluate the gains offered by randomly wired architectures across a wide variety of graph convolutions and tasks, we adopt a recently proposed GNN benchmarking framework (Dwivedi et al., 2020), that has introduced new datasets and allows for reproducible experiments.

Table 1: ZINC Mean Absolute Error.

| | $L = 8$ | $L = 16$ | $L = 32$ |
|---|---|---|---|
| GCN | $0.465^{\pm0.012}$ | $0.445^{\pm0.022}$ | $0.426^{\pm0.011}$ |
| **RAN-GCN** | $\mathbf{0.447}^{\pm0.019}_{1.5\sigma}$ | $\mathbf{0.398}^{\pm0.015}_{2.1\sigma}$ | $\mathbf{0.385}^{\pm0.015}_{3.7\sigma}$ |
| GIN | $0.444^{\pm0.017}$ | $0.461^{\pm0.022}$ | $0.633^{\pm0.089}$ |
| **RAN-GIN** | $\mathbf{0.398}^{\pm0.004}_{2.7\sigma}$ | $\mathbf{0.426}^{\pm0.020}_{1.6\sigma}$ | $\mathbf{0.540}^{\pm0.155}_{1.0\sigma}$ |
| GatedGCN | $0.339^{\pm0.027}$ | $0.284^{\pm0.014}$ | $0.277^{\pm0.025}$ |
| **RAN-GatedGCN** | $\mathbf{0.310}^{\pm0.010}_{1.1\sigma}$ | $\mathbf{0.218}^{\pm0.017}_{4.7\sigma}$ | $\mathbf{0.215}^{\pm0.025}_{2.5\sigma}$ |
| GraphSage | $0.363^{\pm0.005}$ | $0.355^{\pm0.003}$ | $0.351^{\pm0.009}$ |
| **RAN-GraphSage** | $0.368^{\pm0.015}_{1.0\sigma}$ | $\mathbf{0.340}^{\pm0.009}_{5.0\sigma}$ | $\mathbf{0.333}^{\pm0.008}_{2.0\sigma}$ |
| GAT | $0.416^{\pm0.016}$ | $0.384^{\pm0.011}$ | $0.357^{\pm0.011}$ |
| **RAN-GAT** | $0.430^{\pm0.020}_{0.9\sigma}$ | $0.392^{\pm0.012}_{0.7\sigma}$ | $0.368^{\pm0.011}_{1.0\sigma}$ |

Table 2: CLUSTER Accuracy.

| | $L = 8$ | $L = 16$ | $L = 32$ |
|---|---|---|---|
| GCN | $48.71^{\pm3.04}$ | $48.57^{\pm7.85}$ | $55.62^{\pm3.12}$ |
| **RAN-GCN** | $\mathbf{58.61}^{\pm3.15}_{3.3\sigma}$ | $\mathbf{62.24}^{\pm1.64}_{1.7\sigma}$ | $\mathbf{63.32}^{\pm0.99}_{2.5\sigma}$ |
| GIN | $49.93^{\pm1.79}$ | $49.04^{\pm2.51}$ | $44.96^{\pm5.56}$ |
| **RAN-GIN** | $\mathbf{54.38}^{\pm2.52}_{2.5\sigma}$ | $\mathbf{56.58}^{\pm6.26}_{3.0\sigma}$ | $\mathbf{56.19}^{\pm2.91}_{2.0\sigma}$ |
| GatedGCN | $63.10^{\pm2.54}$ | $70.09^{\pm1.89}$ | $71.94^{\pm1.51}$ |
| **RAN-GatedGCN** | $63.85^{\pm2.45}_{0.3\sigma}$ | $\mathbf{72.13}^{\pm1.68}_{1.1\sigma}$ | $\mathbf{74.32}^{\pm0.89}_{1.6\sigma}$ |
| GraphSage | $66.22^{\pm0.73}$ | $71.50^{\pm1.03}$ | $70.23^{\pm1.77}$ |
| **RAN-GraphSage** | $\mathbf{67.21}^{\pm3.23}_{1.4\sigma}$ | $\mathbf{71.90}^{\pm2.09}_{0.4\sigma}$ | $\mathbf{72.56}^{\pm2.08}_{1.3\sigma}$ |
| GAT | $54.35^{\pm4.39}$ | $60.68^{\pm6.10}$ | $55.41^{\pm4.31}$ |
| **RAN-GAT** | $\mathbf{63.38}^{\pm2.49}_{2.1\sigma}$ | $\mathbf{69.68}^{\pm1.58}_{1.5\sigma}$ | $\mathbf{70.93}^{\pm1.18}_{3.6\sigma}$ |

We focus on testing five of the most commonly used graph convolution definitions: GCN (Kipf & Welling, 2017), GIN (Xu et al., 2019)[2], Gated GCN (Bresson & Laurent, 2017), GraphSage (Hamilton et al., 2017), GAT (Veličković et al., 2017). We select three representative tasks introduced by (Dwivedi et al., 2020): graph regression on the ZINC dataset, node classification on the CLUSTER dataset, and graph classification with superpixels on CIFAR10. To ensure reproducibility, we use exactly the same setting as (Dwivedi et al., 2020). We are interested in the performance differences between the baseline ResNet architecture, i.e., a feedforward architecture with skip connections after every layer, and the randomly wired architecture. It was already shown in (Dwivedi et al., 2020) that ResNet GNNs significantly outperform architectures without residual connections. We remark that other works proposed methods to build deeper GNN (Rong et al., 2019; Zhao & Akoglu, 2019; Gong et al., 2020), but such techniques can be regarded as complementary to our work. We do not attempt to optimize a specific method, nor we are interested in comparing one graph convolution to another. A fair comparison is ensured by running both methods with the same number of trainable parameters and with the same hyperparameters. In particular, the learning rate of both methods is adaptively decayed between $10^{-3}$ and $10^{-5}$ by halving according to the value of the validation loss, with a patience of 5 epochs. Stopping criterion is validation loss not improving for 5 epochs after reaching the minimum learning rate. We average the results of all experiments over 4 runs with different weight initializations and different random architecture graphs, drawn with $p = 0.6$. We also evaluate results for multiple values of the total number of layers (architecture nodes) $L$, in order to show that randomly wired GNNs allow a more effective increase in capacity. The random architectures use sequential paths (Sec. 3.2) in the ZINC experiment, sequential paths and DropPath in the CLUSTER experiment, and only DropPath in CIFAR10[3]. The reason for these choices is that the regression task in ZINC and the node classification task in CLUSTER are particularly sensitive to the size of the receptive field, as observed by analyzing the experimental receptive radius (supplementary material). On the other hand, CIFAR10 is bottlenecked by overfitting, and it greatly benefits from the regularizing effect of DropPath, as also observed on CLUSTER. The number of DropPath iterations in testing was fixed to 16.

## 4.1 RANDOM GNN BENCHMARKING

The results presented in this section show that randomly wired GNNs have compelling performance in many regards. First of all, they typically provide higher accuracy or lower error than their ResNet counterparts for the same number of parameters. Moreover, they are more effective at increasing capacity than stacking layers: while they are essentially equivalent to ResNets for very short networks, they enable larger gains when additional layers are introduced.

Table 1 shows the results obtained on the ZINC dataset. The metric is mean absolute error (MAE), so lower is better. The superscript reports the standard deviation among runs and the subscript reports the level of significance by measuring how many baseline standard deviations the average value of the

---

[2]GIN and RAN-GIN compute the output as in (Xu et al., 2018), using the contributions of all architecture nodes.

[3]We do not use DropPath for RAN-GIN on any experiment as we observed unstable behavior.

Table 3: CIFAR10 Accuracy.

|  | $L = 8$ | $L = 16$ | $L = 32$ |
|---|---|---|---|
| GCN | $54.85^{\pm0.20}$ | $54.74^{\pm0.52}$ | $54.76^{\pm0.53}$ |
| **RAN-GCN** | $\mathbf{57.81}^{\pm0.08}_{14.8\sigma}$ | $\mathbf{57.29}^{\pm0.44}_{4.9\sigma}$ | $\mathbf{58.49}^{\pm0.21}_{7.0\sigma}$ |
| GIN | $48.59^{\pm1.60}$ | $47.14^{\pm1.75}$ | $36.90^{\pm4.71}$ |
| **RAN-GIN** | $\mathbf{52.52}^{\pm0.66}_{2.5\sigma}$ | $\mathbf{52.07}^{\pm1.78}_{2.8\sigma}$ | $\mathbf{42.73}^{\pm7.93}_{1.2\sigma}$ |
| GatedGCN | $68.27^{\pm0.80}$ | $69.16^{\pm0.66}$ | $69.46^{\pm0.47}$ |
| **RAN-GatedGCN** | $68.86^{\pm1.64}_{0.7\sigma}$ | $\mathbf{72.00}^{\pm0.44}_{4.3\sigma}$ | $\mathbf{73.50}^{\pm0.68}_{8.6\sigma}$ |
| GraphSage | $65.58^{\pm0.46}$ | $66.12^{\pm0.11}$ | $65.33^{\pm0.34}$ |
| **RAN-GraphSage** | $65.31^{\pm0.38}_{0.6\sigma}$ | $66.10^{\pm1.11}_{0.2\sigma}$ | $\mathbf{67.68}^{\pm0.37}_{6.9\sigma}$ |
| GAT | $64.43^{\pm0.33}$ | $63.61^{\pm0.66}$ | $64.62^{\pm0.65}$ |
| **RAN-GAT** | $\mathbf{66.18}^{\pm0.65}_{5.3\sigma}$ | $\mathbf{66.27}^{\pm0.16}_{4.0\sigma}$ | $\mathbf{66.01}^{\pm0.38}_{2.1\sigma}$ |

Table 4: Median relative gain over $L = 4$.

|  |  | $L = 8$ | $L = 16$ | $L = 32$ |
|---|---|---|---|---|
| ZINC | ResNet | +7.88% | +17.06% | +17.99% |
|  | **Random** | **+14.22%** | **+21.81%** | **+24.36%** |
| CLUSTER | ResNet | +17.90% | +15.80% | +14.26% |
|  | **Random** | **+20.75%** | **+30.07%** | **+32.41%** |
| CIFAR10 | ResNet | −0.84% | −0.14% | −1.22% |
|  | **Random** | **+1.31%** | **+3.58%** | **+4.10%** |

Table 5: Comparison against SIGN and PPNP

|  | Num. param. | GCN (no residuals) | GCN | RAN-GCN | PPNP | SIGN |
|---|---|---|---|---|---|---|
| ZINC | 180k | 0.526 | 0.465 | **0.447** | 0.746 | 0.566 |
| CLUSTER | 180k | 22.23 | 48.71 | **58.61** | 33.00 | 48.35 |
| CIFAR10 | 180k | 51.16 | 54.85 | **57.81** | 36.37 | 52.49 |
| ZINC | 360k | 0.537 | 0.445 | **0.398** | 0.750 | 0.555 |
| CLUSTER | 360k | 19.26 | 48.57 | **62.24** | 37.37 | 48.51 |
| CIFAR10 | 360k | 49.86 | 54.74 | **57.29** | 36.68 | 53.55 |
| ZINC | 720k | 0.649 | 0.426 | **0.385** | 0.804 | 0.574 |
| CLUSTER | 720k | 20.90 | 55.62 | **63.32** | 28.77 | 49.14 |
| CIFAR10 | 720k | 47.47 | 54.76 | **58.49** | 38.54 | 53.72 |

random architecture deviates from the average value of the baseline. Results are in bold if they are at least $1\sigma$ significant. The results show that the randomly wired GNNs typically outperform the ResNet baseline by significant margins. Table 2 reports the node classification accuracy on the CLUSTER dataset. It can be seen that the random architectures achieve very significant improvements on this dataset, especially for RAN-GCN, RAN-GIN and RAN-GAT. Table 3 reports the classification accuracy on CIFAR10 when the images are converted to graphs using superpixels. Also in this case, the randomly wired architecture greatly outperforms the baseline, in some cases achieving gains higher than $5\sigma$. Finally, Table 4 shows the relative performance gain (relative improvement in accuracy or mean absolute error), averaged over all the graph convolution definitions, with respect to a short 4-layer network, where random wiring and ResNets are almost equivalent (results in supplementary material). We can notice that deeper ResNets always provide lower gains with respect to their shallow counterpart than the randomly wired GNNs. Moreover, we observe monotonically increasing gains for random GNNs while deeper ResNets are either unable to significantly extract more performance beyond $L = 16$ or even provide worse results than the $L = 4$ network. This supports our claim that the randomly wired architecture is a superior way to increase GNN capacity.

Finally, we compare the proposed method against two other frameworks for GNNs, namely PPNP Klicpera et al. (2018) and SIGN Rossi et al. (2020), which propose different approaches for solving the oversmoothing problem. Due to the significant differences in the approaches, providing a fair comparison is challenging. We decided to equalize the number of parameters across the methods, since notions as number of layers or features cannot be translated in all the frameworks (180k,360k,720k parameters correspond to the $L = 8, 16, 32$ settings in the previous tables). Table **??** shows the obtained results. We can observe that both PPNP on node classification and SIGN on all tasks outperform the standard GCN architecture without skip connections, but they cannot outperform GCN with residual connections and the randomly wired GCN.

Table 6: Edge probability, $L = 16$, RAN-GCN.

|  | $p = 0.2$ | $p = 0.4$ | $p = 0.6$ | $p = 0.8$ |
|---|---|---|---|---|
| ZINC | $0.440^{\pm 0.025}$ | $0.427^{\pm 0.025}$ | $\mathbf{0.409}^{\pm 0.010}$ | $0.415^{\pm 0.012}$ |
| CLUSTER | $59.87^{\pm 1.64}$ | $60.71^{\pm 2.27}$ | $\mathbf{62.75}^{\pm 2.32}$ | $62.93^{\pm 2.75}$ |
| CIFAR10 | $56.53^{\pm 0.61}$ | $56.21^{\pm 0.48}$ | $\mathbf{57.44}^{\pm 0.46}$ | $56.06^{\pm 0.48}$ |

Table 7: DropPath on CIFAR10, RAN-GatedGCN. No sequential path embedding.

|  | $L = 8$ | $L = 16$ | $L = 32$ |
|---|---|---|---|
| None | $68.07^{\pm 0.94}$ | $70.78^{\pm 0.38}$ | $72.75^{\pm 0.37}$ |
| DropPath | $\mathbf{68.86}^{\pm 1.64}$ | $\mathbf{72.00}^{\pm 0.44}$ | $\mathbf{73.50}^{\pm 0.68}$ |

Table 8: DropPath on CIFAR10, RAN-GatedGCN. No sequential path embedding.

| $p_{\text{drop}}$ | | | | |
|---|---|---|---|---|
| 0 | 0.005 | 0.01 | 0.02 | 0.03 |
| $70.78^{\pm 0.38}$ | $70.90^{\pm 0.46}$ | $\mathbf{72.00}^{\pm 0.44}$ | $71.55^{\pm 0.83}$ | $71.09^{\pm 1.79}$ |

Table 9: Sequential path embedding on CLUSTER, RAN-GatedGCN. No DropPath.

|  | $L = 8$ | $L = 16$ | $L = 32$ |
|---|---|---|---|
| Fully random | $56.93^{\pm 5.17}$ | $66.50^{\pm 5.10}$ | $70.38^{\pm 1.07}$ |
| Random+Sequential | $\mathbf{63.30}^{\pm 2.15}$ | $\mathbf{68.89}^{\pm 1.87}$ | $\mathbf{71.65}^{\pm 0.97}$ |

## 4.2 ABLATION STUDY

### 4.2.1 EDGE PROBABILITY

We first investigate the impact of the probability $p$ of drawing an edge in the random architecture. Table 6 shows the results for a basic random architecture without DropPath nor embedded sequential path. It appears that an optimal value of $p$ exists that maximizes performance. This could be explained by a tradeoff between size of receptive field and the ability to modulate it.

### 4.2.2 DROPPATH

The impact of DropPath on CIFAR10 is shown in Table 7. We found the improvement due to DropPath to be increasingly significant for a higher number of architecture nodes, as expected due to the increased number of edges. The value of the drop probability $p_{\text{drop}} = 0.01$ was not extensively cross-validated. However, Table 8 shows that higher drop rates lowered performance.

### 4.2.3 EMBEDDED SEQUENTIAL PATH

The impact of embedding a sequential path as explained in Sec. 3.2 is shown in Table 9. It can be observed that its effect of promoting receptive fields with larger radius is useful on this task. We remark that, while we do not report results due to space constraints, this is not always the case and some tasks (e.g., CIFAR10) do not benefit from promoting larger receptive fields.

## 5 CONCLUSIONS

We showed how randomly wired architectures can boost the performance of GNNs by merging receptive fields of multiple size. Consistent and statistically significant improvements over a wide range of tasks and graph convolutions suggest considering them as the go-to choice for new models.

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
