# OpenReview forum: "Don't stack layers in graph neural networks, wire them randomly"
_ICLR.cc/2021/Conference — Reject_

### Official Review · AnonReviewer3 · 2020-10-20
**Questionable experiment result with unclear explanation of motivation**

**Rating:** 4
**Confidence:** 4

**Review:**

Summary:
The authors proposed to randomly wire the GNN layers. They claim it can not only resolve over-smoothing problem but also enable the varied size of receptive fields.

Pros:
1.	Really great looking graphics.

Cons:
1.	The motivation of why should we use Erdos-Renyi (ER) graph to generate DAG for the random rewiring is not clear.
2.	How can we learn from the results of the DAG statistics (such as averaged path length) is unclear. I don’t understand how this can help us design or improve the proposed architecture.
3.	The experimental results seem inconsistent to [1], where the authors claim to follow their experiment setting.

Detailed comments:

The main weakness of this paper is its motivation. I do not see any theoretical reasoning that we should use ER graph instead of the other choice of random graph generator. The authors mention that small world and scale-free networks have been studied and use in [2], but I don’t see any detailed comparison why the ER graph is a more preferred choice. Even in the experiment section, I do not aware of any comparison which is unsatisfactory.

The other weakness is that the “theoretical analysis” mentioned in this paper neither leads to any reasoning in model design nor guarantee in performance. What can we learn from knowing Lemma 3.1 and 3.3? How do they explain why using ER graph for random wiring is favorable? I do not even see it helps on choosing the hyperparameter $p$.

The final but most questionable part is the experiment section. The authors claim that they adopt a recently proposed GNN benchmarking framework [1]. However, the reported results are significantly different and much worse than those reported in [1]. For example, GCN ($L=16$) has 68.5% of accuracy in the experiment on CLUSTER dataset in [1] (Table 2, Node classification). In contrast, the authors report GCN accuracy for only 48.57%. Of course, the hyperparameters might be chosen differently from [1], but then the question would be why not optimize it according to [1]? Also, why not report the performance of more shallow $L$ (say $L=2$ or $4$) which is a more common choice? All these inconsistency makes me hard to believe the proposed methodology   would work.

Reference:

[1] “Benchmarking Graph Neural Networks,” Dwivedi et al., arXiv preprint arXiv:2003.00982, 2020.

[2] “Exploring randomly wired neural networks for image recognition.,” Xie et al., In Proceedings of the IEEE International Conference on Computer Vision, pp. 1284–1293, 2019.

---

> ### Author Response · Authors · 2020-11-12
> **Important remarks on the validity of the experiments and paper motivations**
>
> The reviewer is concerned with our use of the Erdos-Renyi (ER) random graph as the DAG in our model. However, we need to remark that the focus of the paper is not which random graph is used for the architecture but the idea of using a random graph itself in the context of graph neural networks. While [2] introduced the notion of random graph architectures for classical CNNs on image classification problems, their use with graph neural networks had not been explored before. In this sense, the motivation of our paper is to study what kind of advantage randomly wired graph neural networks have with respect to the widely used ResNets. Our analysis shows that randomly wired architectures merge the contributions of multiple receptive fields and can even modulate the effective size of the receptive field, which are properties not possessed by ResNets and seem to be very useful on graph problems, leading to better and richer representations. Our analysis and experiments treat the ER random graph as an example of a random architecture, for which precise mathematical properties can be easily derived. However, we do not claim that the ER model is the best among random graph generators. In fact, we expect most of intuition behind the modulation and merging of receptive fields to carry over to scale-free or small-world graphs ([2] showed that all these three graph models performed roughly equally). Further exploration of which DAG generator is best would be interesting but it is outside of the scope of this work, which aims at showing the importance of receptive field manipulation property offered by random wirings.
>
> For what concerns the role of the theoretical analysis, the reviewer is right in saying that it does not provide guarantees on system performance or tells how to choose the optimal value of $p$ or design the architecture optimally for a given problem. However, this was not the objective of the analysis, whose goal is to provide insights on what the randomly wired architecture is doing (this was not explored in [2]). In particular, our main insight is given by lemma 3.2 which allows to understand that the receptive field size can be modulated by the weights over the paths. However, in order to understand how this is happening, lemma 3.1 and 3.3 characterize the path interpretation of the architecture, showing how the receptive field radius is fixed in a ResNet while it can be tuned, either manually by selection of the hyperparameter $p$ and/or automatically by the aggregation weights $\omega$. This allows to understand Fig.3 where different distributions can be achieved. It is true that those insights do not predict which value of $p$ to choose for a given problem, but the point is to show that the random architecture offers more degrees of freedom and increased flexibility to better suit different problems.
>
> The reviewer correctly points out the discrepancy between our results and the results reported in [1]. The reference provided by the reviewer (and the one we incorrectly put in the paper) points to the latest version of the benchmarking framework (v3) available today. However, all our experiments were performed on the v1 version (available at https://arxiv.org/abs/2003.00982v1) and are consistent with that. Later versions were updated simultaneously to our work and we kept v1 for consistency. This does not hinder reproducibility as it just requires to follow the v1 setup. Later settings changed some hyperparameters so, as the reviewer noticed, the results are not comparable. In particular, the mentioned GCN experiment on CLUSTER with L=16 used a much lower number of trainable parameters in the v1 (362787, exactly the same as what we use, see supplementary material, instead of 501687 used in v3). We explicitly kept the same settings as the v1 framework for both random and ResNet exactly in order to avoid biases or cherry picking favourable configurations, and our results match those reported in https://arxiv.org/abs/2003.00982v1.
> About the results on shallow networks: the focus of our work is to study the problem of diminishing model performance as layers are added. For this reason we are more interested in observing perfromance for higher values of L. Nevertheless, the results for L=4 are reported in the supplementary material and they are used to compute Table 4. The L=2 configuration is uninteresting because with such a small number of layers, the ResNet baseline and the proposed random architecture are exactly the same. As a matter of fact, the two architectures also perform very similarly at L=4 but the benefits of random wirings start being evident at L=8 and more, by providing larger and monotonically increasing gains.

---

> > ### Comment · AnonReviewer3 · 2020-11-13
> > **Additional comments**
> >
> > I thank reviewer for the fast response and clear explanation. Regarding the experiments, I agree that the reported results are more consistent to those in https://arxiv.org/abs/2003.00982v1 .  Although I think it is okay to keep the experiment as it is by now, new experiments following the latest version of [1] is encouraged. This is due to the fact that the difference is so large that makes me wonder if the same significant gain can still holds under the latest setting in [1]. Indeed, the authors might not have time to complete the new experiments in this short reviewing period so I will not judge the paper by this point.
> >
> > For the use of ER graph, if the purpose of the paper is to merely show that a random wiring method can have better performance, then the contribution in novelty is limited. Note that the random wiring method has been reported in CNN literature, and the only contribution in novelty of the current paper is to use the same idea in GNN. It is fine if further analysis and understanding is made specifically for GNN, which the authors attempt to do so. However, I doubt this is achieved in this paper.
> >
> > On the theoretical analysis of the current paper, the authors mention in their response that the main implication of their analysis is to show that the random architecture offers more degrees of freedom and increased flexibility to better suit different problems. The point that "random architecture offers more degrees of freedom" is trivial to me, as we all know that there is many more different realization of ER graphs compare to a complete graph. Thus the ER graphs obviously offers more degree of freedom. The real important question to ask is what benefit does this "degrees of freedom" brings to us. In my opinion, the authors merely show that receptive field size will vary which is again a straightforward conclusion. The only contributions for Lemma 3.1 to 3.3 is to show that the weight $w$ and the hyperparameter $p$ can affect the receptive field size in some manner. However, this conclusion does not answer the most important question: "How does the proposed method resolve a problem in existing GNN design?" For example, if the authors want to show that the random rewiring can really resolve the over-smoothing problem, they should give a theoretical analysis to prove that. Note that there are already many works resolve the over-smoothing problem "theoretically" (i.e. [3] as cited by the authors), why should we favor the current random wiring procedure without theoretical guarantee? On the other hand, the authors does not show how randomly varying the receptive field size is more favorable compare to Resnet. Indeed, there is a "optimal" computational graph (the way we wire layers) and it is very likely not to be the complete graph. However, why a random graph is better or closer to this optimal graph? I think these are the real questions to ask and analyze for the proposed methodology.
> >
> > In summary, I'm still not convinced that the current paper should be accepted, as its contributions in theory and novelty are limited. Although their empirical evaluation shows some benefit of random wiring procedure over ResNet, I feel like more study is needed to validate the methodology. I kind of agree now that the experiments are reproducible, hence I tend to slightly increase my score to 4.
> >
> > References
> >
> > [3] Kenta Oono and Taiji Suzuki.  Graph neural networks exponentially lose expressive power for node classification. In International Conference on Learning Representations, 2020.

---

> > > ### Author Response · Authors · 2020-11-18
> > > **Re: Additional comments**
> > >
> > > As we understand it, the reviewer is mainly concerned with having a rigorous proof of the benefits of randomly wired architectures in order to answer the question "How does the proposed method resolve a problem in existing GNN design?". As is often the case for deep learning methods, providing a rigourous proof of the exact benefits could be quite hard, and it is not in the scope of our contribution. The reference [3] itself provided by the reviewer is merely a new important piece of knowledge but does not conclusively "resolve the over-smoothing problem theoretically", especially because the analysis in [3] does not explain the better performance of residual networks (the authors themselves discuss in the paper that there must be something more to it). However, our theoretical justifications are not without merits as they fit a line of reasoning that stretches several works in the GNN literature. We can schematize it in the following way:
> > > - several works in the GNN literature (Jumping Knowledge Networks, SIGN, ...) pointed out that merging contributions of receptive fields of multiple size is beneficial;
> > > - we propose randomly wired GNNs by extending the framework developed for CNN and observe strong experimental results;
> > > - we develop an analysis (novel to this work, not present in the original paper on randomly wired CNNs) of what these architectures are doing that shows that, when viewed as ensembles of paths, they can merge contributions of multiple receptive fields, control them with more degrees of freedom of existing architectures and even adapt their size dynamically thanks to the trainable weights on the paths;
> > > - this analysis shows that randomly wired GNNs can implement the intuition of the aforementioned GNN works in a highly general and tunable fashion, nicely fitting the existing literature and expanding it. We do not think that the insights and connections with the GNN literature we provide are "trivial".
> > >
> > > Besides theoretical insights, we believe our contribution should be not disregarded as it shows strong experimental improvements in a highly reproducible setting. These strong improvements were not observed on classic CNNs where the randomly wired architecture basically performed as well as other baselines, highlighting the uniqueness of the graph setting. We also remark that DropPath and Sequential path embeddings are entirely novel contributions that further improve the performance and based on theoretical insights (decorrelating path contributions and promoting larger receptive fields, respectively).

---

### Official Review · AnonReviewer1 · 2020-10-26

**Rating:** 5
**Confidence:** 3

**Review:**

Summary:

This paper extends the technique of randomly wired neural nets from [1] to Graph Neural Networks and show that they perform better than tradtional GNN architectures. They demonstrate the improved capacity of this architecture via a number of experiments on the benchmark in [2] and ablation studies.

Reason for score:

Pros:

  - The paper evaluates the technique on a widely accepted benchmark for Graph Neural Networks.
  - Diminishing model performance in GNNs with increased number of layers is an important problem and it looks like randomly wired GNNs provide monotonically increasing performance for up to  L = 32
  - The models generated using random wiring outperform the baseline model across a large number of settings such as the graph convolution operator used and the number of layers
  - Augmenting the randomly wired networks with a sequential path is interesting.
  - Ablation experiments are extensive and convincing.

Cons:

  - The novelty is only incremental, building on the core idea from [1], but the results are strong so this is not a big issue.
  - Why didn't the paper test on other tasks from the benchmark in [2] like PATTERN and TSP? A full set of experiments would rule out the possibility that the benchmark tasks were cherry picked.

Overall:

  Extension of an existing method to GNNs which produces strong results. I vote to accept this paper.

Questions:

 - How many iterations of inference do you'll do for MonteCarlo DropPath during testing?


References:

[1] Saining Xie, Alexander Kirillov, Ross Girshick, and Kaiming He. Exploring randomly wired neuralnetworks for image recognition. In Proceedings of the IEEE International Conference on ComputerVision, pp. 1284–1293, 2019.

[2] Vijay Prakash Dwivedi, Chaitanya K Joshi, Thomas Laurent, Yoshua Bengio, and Xavier Bresson.Benchmarking graph neural networks. arXiv preprint arXiv:2003.00982, 2020.

---

> ### Author Response · Authors · 2020-11-12
> **Some clarifications**
>
> We thank the reviewer for their comments. While novelty might seem incremental, beyond the strong results, we also show an analysis of what random wirings are doing by viewing them as emsembles of paths. This analysis is novel as it was not presented in [1].
>
> Concerning the choice of datasets, we made sure to choose one dataset per task (ZINC=graph regression, CLUSTER=node classification, CIFAR=graph classification) to maximize the variety of tasks given the space constraints. Some datasets such as TSP also are very computationally-demanding so it would have been difficult to provide statistically reliable results with a large number of runs and configurations.
>
> We used 16 iterations of MonteCarlo DropPath in testing. We clarified this point in the revised text.

---

> ### Comment · Area_Chair1 · 2020-11-18
> **Author response**
>
> Dear AnonReviewer1,
>
> We are now entering the second discussion stage. Could you please check whether the authors have addressed your concerns and questions and potentially ask any further clarification questions?
>
> Thank you,
> Your Area Chair

---

### Official Review · AnonReviewer4 · 2020-10-27
**A nice paper about a randomly-wired GCNs**

**Rating:** 8
**Confidence:** 4

**Review:**

Summary:

The paper proposes a new method for building graph convolutional neural networks. It shows, that during the building of the network, instead of stacking many layers and adding the residual connection between them, one could employ a randomly-wired architecture, that can be a more effective way to increase the capacity of the network and thus it could obtain richer representations. The proposed method is an interesting direction in the field of graph convolutional neural networks. The new method could be seen asa generalization of the residual networks and the jumping knowledge networks.

=============================================================================

Pros:

1. The paper proposes a novel, randomly-wired architecture for building the graph convolutional neural networks. Moreover authors analyze proposed randomly-wired architectures and show that they are generalizations of ResNets.

2. The authors provide the theorethical analysis of the radius of te receptive field of GCN. They show that by using randomly-wired network, together with trainable weights on the architecture edges and sequential path, the network could tune the desired size of the receptive fields to be merged to achieve an optimal configuration for the problem.

3. The authors propose the MonteCarlo DropPath regularization - a novel regularization method for randomly-wired architectures, that is related to dropout, however is carried out at a higher level of abstraction.

4. The authors provide a comprehensive experimental results of the proposed method - they compare various GCN architectures, created on traditional and randomly-wired way, on three representative tasks - graph regression, graph classification and node classification. Moreover they show, that randomly-wired GCNs gets better results than ResNet GCNs on almost all tested cases. Moreover the authors shows that deeper randomly-wired GCNs always provide bigger gains with respect to their shallow counterpart than ResNet GCNs.

=============================================================================

Cons:

1. Figure 1 is not clear to me. I am not sure what the colored point cloud is about. The authors should consider rewriting a description of this figure.

2. In the final version of the paper, the ablation study should be reported on all datasets (however the authors remark that, they do not report results on this version of paper due to space constraints).

3. I would like to see the more extensive analysis of DropPath, e.g what are the scores for different levels of the drop probability.

=============================================================================

Questions during rebuttal period:

1. Why the authors use different types of GCNs during the ablation study?

=============================================================================

=============================================================================

Reasons for score:

Overall, I vote for accepting this paper. The idea proposed by the authors is novel and confirmed theoretically and experimentally.
My major concern is about ablation study and the clarity of one figure. Hopefully the authors can address my concern in the rebuttal period.

---

> ### Author Response · Authors · 2020-11-12
> **A few remarks on the points raised by the reviewer**
>
> We thank the reviewer for the useful comments. We have improved the caption of Fig. 1, better describing the meaning of the colored point clouds. This is a representation of the input graph, where the colors describe the receptive field.
>
> Before the end of the discussion period we will upload a new version of the paper with an extended ablation study. We will add an ablation study on the drop probability of DropPath. Moreover, in Table 5 we will add the results on CLUSTER, so that this table will report the results on all the datasets that we have considered. Instead, Table 6 and 7 analyze the contribution of DropPath and sequential path. As explained in the experimental setting, these design choices are not always used, because they have an impact only on some datasets. Therefore, in the ablation study we just focus on those datasets where they are actually used.
> In the ablation study we decided to use the Gated-GCN because this definition of graph convolution is the one that shows the best performance on the datasets that we considered. Since these experiments are aimed to assess the performance of the proposed design choices, we are interested in evaluating if they truly hold on the highest performing definition of graph convolution to avoid other potential confounding factors. Instead, the experiment reported in Table 5 just shows an example of the influence of the $p$ value on the performance. We are not interested in finding the optimal value of p, but we just want to show that usually there is a trend where we can identify an optimal value of p. For this reason, we decided to use a very popular and low-complexity definition, namely the GCN.

---

> > ### Author Response · Authors · 2020-11-18
> > **Updated paper**
> >
> > We updated the paper to expand the ablation experiments, by including the missing CLUSTER experiment and the ablation of the $p_{drop}$ value.

---

> ### Comment · Area_Chair1 · 2020-11-18
> **Author response**
>
> Dear AnonReviewer4,
>
> We are now entering the second discussion stage. Could you please check whether the authors have addressed your concerns and questions and potentially ask any further clarification questions?
>
> Thank you,
> Your Area Chair

---

### Official Review · AnonReviewer2 · 2020-11-02
**More details are needed.**

**Rating:** 5
**Confidence:** 4

**Review:**

This paper utilizes Randomly Wired architectures to boost deep GNNs. Theoretical analyses verify that randomly wired architectures behave like path ensemble and it enables adaptive receptive field. Experimental results on three non-popular datasets demonstrate the strength of the proposed model. Overall, the idea is interesting. Yet this paper can be made better through the following aspects:

1. This paper contains confusing equations and notations. For example, in Eq 1., why w_ij equals \sigma(w_ij). What's the meaning of domain nodes and how do we connect the architecture nodes and the domain nodes. What's the definition of \mathcal{A}?

2. This paper only proposes the recursion formula but omits some basic definitions, i.e. the definition of h^{(i)}. Where does the recursion start?  Is there an initialized h^{(0)}?

3. The algorithm framework is not clearly depicted. How do R-GCNs accomplish the graph propagation process?

4. Insufficient experimental comparisons. How do R-GCNs and GCNs perform when L=2. How do R-GCNs perform on standard node classification datasets, such as Cora, Citeseer, and Reddit, since deep GCNs fail particularly on node  classificaiton. How do R-GCNs perform against other deep frameworks such as APPNP and JKNet, both of which resort to more sophisticated skip connections than ResGCN.

#############post-rebuttal############

I have carefully checked all other reviewers' comments, the authors' response, and the revised version. Thank the authors for their detailed feedback. They have addressed my concerns on the unclear presentation. However, joining the comments from other reviewers (particularly R3), I still think there are two major issues that prevent me from further increasing my score.

Q1. It is still unclear why the proposed model can tackle the over-smoothing issue in existing deep GCNs.

This paper has theoretically revealed the benefit of adaptive ensemble paths towards better trainability. Given the claim in Introduction, it is still unclear why such benefit can be used to relieve over-smoothing, particularly due to the missing analysis of the output dynamics. As already pointed out by R3, [3] has set up a nice notion of framework on explaining how over-smoothing happens and why deep GCN fails. It is a pity that this paper has not put their analyses into this framework and discussed the relation with the over-smoothing issue. Actually, a more in-depth discussion of over-smoothing on general GCNs (including ResGCN, APPNP) has also provided in an arXiv preprint paper [4]. It does show that the residual networks are capable of slowing down the convergence speed to the subspace and thus alleviating over-smoothing. Since the idea of random wiring is initially proposed in CNNs, the contribution of this paper that we expect is to answer how this idea can be utilized to solve the specific weakness in the graph domain.

Q2. The experimental evaluations are still unconvincing.

It is thankful that the authors have additionally provided the performance of SIGN and APPNP in the revised version. Yet, the reported accuracies of APPNP seem weird and much worse than other baselines. I do not agree with the authors' response that APPNP is not intended to address the over-smoothing problem. As experimentally shown in [5] and theoretically analyzed in [4], keeping the connection between each middle layer and the input layer is able to prevent the output from converging to the subspace caused by over-smoothing, and thereby deliver desired performance with the increase of depth. As this paper has conducted experiments on a newly-public benchmark under inconsistent experimental setting up (raised by R3), it is hard to justify the significance of the proposed idea compared with previous methods, specifically given the irrational observations on APPNP.

Hence, I still believe this paper is below the acceptance line.

[3] Kenta Oono and Taiji Suzuki. Graph neural networks exponentially lose expressive power for node classification. In International Conference on Learning Representations, 2020.
[4] Tackling Over-Smoothing for General Graph Convolutional Networks, arXiv 2020. [5] Simple and Deep Graph Convolutional Networks, NIPS 2020.

---

> ### Author Response · Authors · 2020-11-12
> **Clarifications on details**
>
> [Response split in two posts: 1/2]
> We thank the reviewer for the useful comments. We apologize if the notation was confusing, we improved it in the revised version of the paper. The operation described in Eq. (1) is the same as the one presented in (Xie et al., 2019). We refer to this paper for a deeper discussion on this definition, but it seems that constraining the range of those aggregation weights with a sigmoid function improves training stability. In the revised paper, we have clarified the meaning of domain nodes, better highlighting the difference with respect to the architecture nodes. A GNN takes as input a graph where the data for that specific problem live and we define as domain nodes the nodes of such graph. Instead, we define as architecture nodes the nodes of the random DAG that describes the architecture of the GNN. Thus, an architecture node is a layer of the randomly-wired GNN. The symbol $\mathcal{A}_i$ is defined right after Eq. (1) and represents the set of the architecture nodes that are direct predecessors of the architecture node i. Given a DAG with edge set $\mathcal{E}$, we define as direct predecessor of node i all the DAG nodes j such that (j,i) is in $\mathcal{E}$.
>
> Eq. (1) defines the input of the i-th architecture node. In the revised paper, we have clarified the definition of i in Eq. (1). In a randomly wired neural network, each architecture node aggregates the outputs of the predecessor nodes, propagating data through the DAG towards the output. This means that at each architecture node the receptive fields of varied size are merged, resulting in richer representations and a mitigation of the depth problem. Figure 1 illustrates this concept. The theoretical analysis presented in Sec. 3 shows that randomly wired neural networks can be seen as an ensemble of paths of varied size and the resulting receptive field is a combination of the receptive fields of shallower networks, induced by each of the paths. Moreover, thanks to the trainable weights $w_{ij}$, the contribution of the various path lengths can be modulated, enabling adaptive receptive fields.
>
> In the experimental validation, we do not show the case when L=2, because in this case, since there are only 2 architecture nodes, there is no difference between the randomly wired GNN and the standard GNN. Moreover, in this paper we are interested in studying the behaviour of GNNs when the number of layers increases, showing that employing a randomly wired architecture can be more effective than building deeper networks by stacking many layers. Therefore, the behaviour of the proposed method with a low number of nodes is of minor importance.
>
> The datasets cited by the reviewer were commonly used in the past for experimental evaluation of GNNs. However, recently many researchers avoid using such datasets since they are too simple, resulting in misleading results and are affected by a high variability across splits (see e.g., (Vignac et al.,2020)). This makes them unreliable. For these reasons, recently some new benchmark datasets have been proposed. In this paper, we decided to evaluate the proposed method on the benchmark proposed in (Dwidedi et al, 2020). The three datasets considered in the experimental section correspond to three different tasks: graph regression on the ZINC dataset, node classification on the CLUSTER dataset, and graph classification on the CIFAR10 dataset. Therefore, the performance of the proposed method on node classification are evaluated using the CLUSTER dataset. In the supplementary material, we show the results on a more classical dataset, i.e., TU ENZYMES. The reported results clearly highlight the limitations of such type of datasets, showing a very high variability and thus impossibility of defining any ranking among methods.

---

> > ### Author Response · Authors · 2020-11-12
> > **Clarifications on details (part 2)**
> >
> > [Response split in two posts: 2/2]
> > Concerning the comparisons, we argue that the methods listed by the reviewer cannot be fairly compared against our framework, either because they are complementary or because they define a completely different approach to constructing graph neural networks and not just an architecture that can be applied to the same graph convolutions considered in our paper.
> > As discussed in the introduction of the paper, the proposed method can be seen as a generalization of the Jumping Network proposed in (Xu et al., 2018), where layer outputs can not only jump to the network output but to other layers as well, continuosly merging receptive fields. We argue that Jumping Networks are not strictly an alternative baseline to the proposed approach as they can be combined. In fact, both the GIN and R-GIN used in the experimental evaluation compute the output as in (Xu et al., 2018), using the contributions of all the architecture nodes. We chose to use this method only for (R-)GIN because GIN is presented in this way in the original paper. In the experiments, we show that R-GIN significantly outperforms the GIN baseline. This means that the randomly wired architecture can provide a performance gain even when we consider a JK network. Instead, the second method cited by the reviewer, namely APPNP, is a different framework for defining a GNN, which redefines both architecture and propagation operations. We do not show a comparison against this method, because it does not fit the problem of defining a general architecture for various definitions of graph convolution that we explore in this paper. In fact, it would not be clear how to fairly compare the proposed approach with APPNP or how to decouple the merits of the proposed architecture from the different graph convolution / propagation mechanisms.

---

> > > ### Author Response · Authors · 2020-11-18
> > > **New experiments**
> > >
> > > We uploaded a revised version of the paper where we added some ablation experiments as requested by AnonReviewer4 and a comparison with the suggested SIGN and APPNP.
> > >
> > > We would like to remark that these two methods are not intended to address the oversmoothing problem while keeping the same framework of the graph convolution definitions we analyzed, but rather novel ways of defining graph neural networks, which also present important advantages such as scaling to large graph sizes. Moreover, in the original paper, APPNP is tested only for semi-supervised node classification and so it may not generalize well to the graph regression or classification we test. This can explain why APPNP often exhibits poor performance in our experiments.
> > > Due to the different approach to GNN definition, providing a fair comparison is also challenging and we decided to equalize the number of trainable parameters across the methods, as notions about the number of layers or features cannot be directly translated. We chose to report only GCN variants (GCN without skips, GCN resnet and RAN-GCN) in the Table as they use the most similar propagation mechanism (isotropic) to APPNP and SIGN. Results for the resnet and random architectures for the other graph convolutions can be directly read from the other tables.
> > >
> > > Overall, APPNP on node classification and SIGN on all tasks typically outperform the standard GCN architecture without skip connections, thus somewhat reducing oversmoothing, but cannot outperform the ResNet variant of GCN, even though SIGN gets close, and definitely not RAN-GCN.

---

> ### Comment · Area_Chair1 · 2020-11-18
> **Author response**
>
> Dear AnonReviewer2,
>
> We are now entering the second discussion stage. Could you please check whether the authors have addressed your concerns and questions and potentially ask any further clarification questions?
>
> Thank you,
> Your Area Chair

---

### Comment · Area_Chair1 · 2020-11-11
**Question about baselines**

To get the discussion started, I would like to extend on R2's question on related baselines:

How does the proposed approach compare to related techniques that are aimed at mitigating the oversmoothing problem in GNNs, namely JK networks [1], PPNP [2], and SIGN [3]?

One minor comment on model naming: R-GCN typically refers to [4]. Maybe it makes sense to choose a different model name to avoid confusion.

[1] Xu et al., Representation Learning on Graphs with Jumping Knowledge Networks (ICML 2018) -- https://arxiv.org/abs/1806.03536
[2] Klicpera et al., Predict then Propagate: Graph Neural Networks meet Personalized PageRank (ICLR 2019) -- https://arxiv.org/abs/1810.05997
[3] Frasca et al., SIGN: Scalable Inception Graph Neural Networks (2020) -- https://arxiv.org/abs/2004.11198
[4] Schlichtkrull et al., Modeling Relational Data with Graph Convolutional Networks (ESWC 2018) -- https://arxiv.org/abs/1703.06103

---

> ### Author Response · Authors · 2020-11-12
> **Addressing reviewers concerns**
>
> We would like to thank all the reviewers for their work. In our posts, we provide a detailed point by point response to their comments. We also uploaded a first revision of the paper that addresses some of the comments. We will upload a further revision with additional ablation experiments as requested by reviewer 4, as soon they are ready.
>
> Regarding the comments raised by the AC, we thank you for pointing out the possible source of confusion on the R-GCN name. We modified the shorthand for all the random architectures into RAN-* to avoid confusion.
>
> Concerning the comparisons, Jumping Networks [1] can be regarded as complementary to our work instead of an alternative as it can be applied on top of randomly wired GNNs as we have done in our GIN and R-GIN experiments (the original presentation of the GIN computed the output in the JK fashion so we kept it). The fact that the R-GIN shows a significant improvement with respect to the GIN baseline proves that the use of a randomly wired architecture can provide an extra performance gain even when we consider a JK network.
> Concerning [2] and [3], we think that it is difficult to perform fair comparisons with those techniques. PPNP and SIGN, more than addressing the oversmoothing problem by modifying the architecture of existing graph neural networks, follow a different approach. Essentially, they propose new ways to construct GNNs including both new architectures and rules for propagation that replace the popular graph convolutions we use in this paper. For example, SIGN does not have layers in the traditional sense as it can be regarded as a single layer with multiple aggregations in parallel. The framework is therefore quite different and even testing PPNP or SIGN on the same data with a similar number of parameters would not be a very informative experiment because it wouldn't tell us whether any difference in the results in favor or against is due to the architecture or the different propagation rules that are used instead of the GCN,GIN,GatedGCN,GraphSage,GAT. To fairly assess the contribution of the randomly wired architecture, we think that only ResNets are a true alternative baseline to the proposed approach because it is the only method that defined a general architecture for the various graph convolutions.

---

> > ### Comment · Area_Chair1 · 2020-11-13
> > **Re: Addressing reviewers concerns**
> >
> > Thank you for your detailed reply.
> >
> > I agree with your sentiment that a direct, fair comparison of your method with other approaches such as PPNP and SIGN is not straightforward.
> >
> > Nonetheless, I would encourage you to consider the value such a comparison would provide to the community and the readers of your paper: we, as a community, want to avoid a situation where we have multiple proposed approaches that address a single underlying phenomenon (loss of information during message passing/diffusion over large receptive fields) that are difficult to compare against one another because many papers decide not to report these comparisons for a variety of reasons. Providing SIGN and PPNP results in the same table, despite these approaches being largely orthogonal to your proposed method, would likely be a helpful signal for the readers of your paper.

---

> > > ### Author Response · Authors · 2020-11-18
> > > **New experiments**
> > >
> > > We thank the Area Chair for this comment. We uploaded a revised version of the paper where we added some ablation experiments as requested by AnonReviewer4 and a comparison with the suggested SIGN and APPNP.
> > >
> > > We would like to remark that these two methods are not intended to address the oversmoothing problem while keeping the same framework of the graph convolution definitions we analyzed, but rather novel ways of defining graph neural networks, which also present important advantages such as scaling to large graph sizes. Moreover, in the original paper, APPNP is tested only for semi-supervised node classification and so it may not generalize well to the graph regression or classification we test. This can explain why APPNP often exhibits poor performance in our experiments.
> > > Due to the different approach to GNN definition, providing a fair comparison is also challenging and we decided to equalize the number of trainable parameters across the methods, as notions about the number of layers or features cannot be directly translated. We chose to report only GCN variants (GCN without skips, GCN resnet and RAN-GCN) in the Table as they use the most similar propagation mechanism (isotropic) to APPNP and SIGN. Results for the resnet and random architectures for the other graph convolutions can be directly read from the other tables.
> > >
> > > Overall, APPNP on node classification and SIGN on all tasks typically outperform the standard GCN architecture without skip connections, thus somewhat reducing oversmoothing, but cannot outperform the ResNet variant of GCN, even though SIGN gets close, and definitely not RAN-GCN.

---

> > > > ### Comment · Area_Chair1 · 2020-11-18
> > > > **Re: New experiments**
> > > >
> > > > Thank you for your swift reply and for running these additional experiments, this is greatly appreciated. I am sure the reviewers will find these additional results helpful.

---

### Decision · Program_Chairs · 2021-01-07
**Final Decision**

**Decision:**

Reject

**Comment:**

This paper proposes to use randomly wired architectures [1] in the context of GNNs and introduces a method for sampling random architectures based on the Erdős–Rényi model. The authors further include a theoretical analysis and two methodological contributions: sequential path embeddings and DropPath, a regularizer. Results are reported on two graph datasets (ZINC and CLUSTER) and on GNN-based CIFAR10 image classification.

The reviewers agree that the empirical results presented in the paper are compelling. The value of the contribution largely lies in this aspect, namely the empirical analysis of an existing technique (randomly wired architectures) in the context of GNNs, in addition to several smaller empirical methodological contributions. I agree with the reviewers in that the nature of the contribution and the otherwise limited novelty calls for a more extensive and detailed empirical evaluation (ideally incl. e.g. FLOPS, wall-clock time, memory usage) across a wide range of datasets and careful ablation studies, and I encourage the authors to improve on this aspect in a future version of the paper. The theoretical analysis is interesting, but, as pointed out by the reviewers both during the reviews and the later discussion period, does not add sufficient value to the main empirical contribution of the paper to push the paper beyond the acceptance threshold and does not satisfactorily address the question of how the method addresses the oversmoothing problem in GNNs.

[1] Xie et al., Exploring randomly wired neural networks for image recognition (ICCV 2019)